# Partial-Failure Segregated Spectrum Assignment for Multicast Traffic in Flex-Grid Optical Networks

Yang Qiu

Key Laboratory of Electronic and Information Engineering (Southwest Minzu University), State Ethnic Affairs Commission, College of Electronic and Information, Southwest Minzu University, Chengdu 610041, China; yqiu@swun.edu.cn

**Abstract:** In this paper, we propose a new algorithm called the partial-failure segregated multicasting routing and spectrum assignment (PFS MRSA) algorithm to improve the service blocking performance of the multicast transmission in flex-grid optical networks (FGONs). By segregating one failure destination leaf-node from a blocked multicast request and accommodating the failure destination leaf-node and the remaining multicast request independently, the success probability of accommodating the originally blocked multicast request can be greatly increased. In this way, the proposed PFS MRSA algorithm can effectively reduce the service blocking probability for the multicast services in FGONs. Simulation results show that the proposed PFS MRSA algorithm achieves significant reduction in service blocking probability when compared with the conventional MRSA algorithms, and such reduction can even reach 100% in some scenarios with low traffic load.

**Keywords:** flex-grid; partial-failure segregated; multicast





## 1. Introduction

In recent years, the intensive growing demands for high-speed services have stimulated the application of wavelength-division multiplexing (WDM) technology in optical networks, which were known as WDM optical networks [1]. By delivering services via their respective designated wavelengths employing arrayed waveguide grating (AWG), conventional WDM optical networks gained the characteristics of optical circuit switching technique, e.g., consistent bandwidth, order-guaranteed end-to-end delivery [2]. These characteristics were especially suitable for stable traffic requests, but lacked flexibility and intelligence when faced with bursty traffic requests [3]. In order to improve the flexibility and intelligence in processing bursty traffic requests, optical packet switching technology was introduced into WDM optical networks by employing the output buffering technique [4], recirculation buffering technique [5], or shared tunable wavelength converters [6]. In addition, some hybrid packet/circuit switching nodes based on AWGs were designed to realize both packet and circuit switching in optical networks [7].

Although the above optical switching technologies can be adopted by WDM optical networks to improve their flexibility and intelligence when faced with bursty services, they still had obvious limitations in provisioning heterogeneous services with diverse bandwidth requirements. Since WDM optical networks assigned constant spectral band (e.g., 50 GHz), also known as wavelength, to different services regardless of their respective bandwidth requirements, their spectral allocation flexibility and utilization efficiency were always deteriorated. Although multi-granularity grooming technology was introduced into WDM optical networks to enhance spectral allocation flexibility and utilization efficiency, its spectrum granularity may limit its flexibility [8]. So as to overcome the limitations of WDM optical networks in spectral allocation and utilization, flex-grid optical networks, also known as elastic optical networks, have been proposed [9–14]. By employing finer allocation granularities (i.e., frequency slots) and more tactful routing and spectral allocation

algorithms, flex-grid optical networks (FGONs) assigned spectral resources to diverse services dynamically according to their respective bandwidth demands. In this way, FGONs greatly improved the spectral allocation flexibility and utilization efficiency.

However, the architectures of FGONs are inherited from traditional WDM networks, which may limit the efficiency of FGONs in accommodating multicast services. Such architectures deliver services via dedicated spectral bands and lack mechanisms to take the advantage of the resource-sharing characteristics in multicast transmission. In addition to the architectures, the spectrum continuity and contiguity constraints in resource allocation may further affect the efficiency of multicast transmission in FGONs. However, the prevalence of some novel point-to-multi-point services, e.g., interactive distance learning and high-definition IP-TV, make efficient multicast transmission highly desirable in FGONs. In this paper, we focus on how to reduce the service blocking probability of FGONs in accommodating multicast services, so that more efficient multicast transmission can be realized in FGONs.

## 2. Related Work

As for FGONs, sophisticated routing and resource assignment (RSA) algorithms are essential to realize their high spectral allocation flexibility and utilization efficiency. Unlike the routing and wavelength assignment (RWA) algorithms adopted in WDM networks [15–17] which seek paths and assign wavelengths for services, RSA algorithms seek paths for service requests and assign an optimized number of contiguous frequency slots (FSs) to them [9,10]. By employing the finer spectral allocation granularity, FS (e.g., 12.5 GHz), and allowing a different number of contiguous FSs assigned to diverse services, RSA algorithms exhibit higher flexibility and require more management than RWA algorithms. Many RSA algorithms, such as the shortest-routing first-fit RSA algorithm [18], distance-adaptive RSA algorithms [19,20], and traffic-split RSA algorithms [21,22], have been proposed for FGONs. In [23–26], some defragmentation-enabled RSA algorithms, employing the make-before-break rerouting technique [23], spectrum-retuning technique [24], independent-sets maximizing technique [25], and push–pull technique [26], were proposed to reduce the generated spectral fragments in FGONs. These defragmentation-enabled RSA algorithms improved the service blocking performance of FGONs, but might induce traffic disruption or extra system complexity. In order to suppress spectral fragments with low system complexity, group-based RSA algorithms were proposed [27–29]. All the above RSA algorithms were proposed for the unicast (i.e., point-to-point) transmission in FGONs, and lacked efficient mechanisms to accommodate multicast transmission in FGONs.

In order to provision multicast transmission with high efficiency in FGONs, experienced multicasting routing and resource allocation algorithms (MRSA), which seek light-trees and assign optimized spectral resources for multicast service requests, are considered as one of the essential enabling technologies [30]. Due to the high expense and complexity required by spectrum conversion technology, the spectral resources allocated to a certain multicast service request by MRSA algorithms should be contiguous and kept unchanged along the sought light-tree for the multicast request until it expires. These spectrum allocation constraints, known as spectrum contiguity and continuity constraints, make designing efficient MRSA algorithms for multicast transmission in FGONs more challenging than designing RWA algorithms for multicast transmission in traditional WDM optical networks [31]. In addition, these two spectrum constraints in MRSA algorithms may induce spectrum fragments into FGONs, while the accumulation of spectrum fragments may deplete the available spectrum resources in FGONs and thus worsen their performance [32,33]. Recently, much research efforts have been performed on designing MRSA algorithms to achieve high-effective multicast transmission in FGONs. In [30], shortest-path-tree (SPT)- and minimum-spanning-tree (MST)- based routing strategies and a first–first spectrum allocation mechanism were introduced into MRSA algorithms. These algorithms optimized the routing paths and allocated resources for multicast transmission in FGONs, so as to improve their efficiency in resource utilization. Noticeably, as SPT and MST are two

typical routing strategies to seek light-trees for multicast services, we adopt these SPT- and MST-based MRSA algorithms [30] as benchmark solutions to MRSA algorithms in this paper. We employ these two MRSA algorithms, namely SPT MRSA and MST MRSA algorithms, in the comparison with the proposed PFS MRSA algorithm. In [34–37], the MRSA algorithms employing a distance-adaptive modulation technique were proposed for FGONs, either with multicast-capable switches [34–36], or with multicast-incapable switches [37]. By applying the adaptive modulation formats for a multicast request based on its transmission distance, the modulation formats with low spectrum efficiency were selected for long-range transmission to guarantee performance, while the modulation formats with higher spectrum efficiency were selected for short-range transmission to increase the resource utilization efficiency. Although the distance-adaptive modulation technique improved the flexibility and efficiency of the MRSA algorithms in spectrum allocation, it usually needed high-quality components or advanced digital signal processing to realize the adaptive modulation format selection [38,39]. In [40–43], light-forest and sub-tree schemes were introduced into MRSA algorithms to improve the flexibility and efficiency in resource allocation. In these MRSA algorithms, by sorting the destination nodes of a multicast request into different groups and seeking the light-trees for each group respectively, sub-trees were accordingly constructed for the multicast request, and each sub-tree can require a dedicated modulation format. In this way, the flexibility in both path routing and spectrum allocation procedures can be improved, which helps increase the efficiency of resource utilization. In [44,45], resource-partition [44], and time-awareness [45] techniques were adopted by MRSA algorithms to minimize the spectral resources consumed in provisioning multicast services. In addition, some other MRSA algorithms were proposed for not only efficient but also survivable multicast transmission in FGONs [46–48]. Although all the above-mentioned MRSA algorithms put efforts on optimizing spectral resources for multicast requests to increase the resource utilization efficiency, they neglected to investigate the diverse influence of each destination in one multicast request, which may greatly affect the service blocking performance of MRSA algorithms, and thus reduced the provisioning efficiency of FGONs. Such provisioning efficiency, defined as the ratio of the successfully provisioned services to all the generated ones, can be improved by reducing service blocking probability. We are focusing on minimizing service blocking probability, $P_b$, to improve provisioning efficiency in this paper, which can be formulated as follows [49]:

$$Minimize \quad P_b = \lim_{T \to \infty} \frac{N_b(T)}{N_t(T)} \qquad (1)$$

where $N_b(T)$ and $N_t(T)$ represent the amounts of the blocked and the total generated multicast service requests in the time duration $[0, T]$, respectively. As for the above-analyzed MRSA algorithms, even when a single destination node failed in the path routing and resource allocation procedure, the whole multicast request could not be provisioned with success. In this way, the service blocking performance of the above-mentioned MRSA algorithms was limited and thus the efficiency of FGONs in provisioning multicast transmission was affected.

In order to increase the provisioning efficiency of FGONs, we propose a partial-failure segregated multicasting routing and spectrum assignment (PFS MRSA) algorithm to improve service blocking performance in this paper. Different from the aboveconventional MRSA algorithms, in which even one failure destination node will result in a multicast accommodation failure, the proposed PFS MRSA algorithm segregates one failure destination leaf-node from a blocked multicast request before accommodating the failure destination leaf-node and the remaining multicast request independently. In this way, the accommodation for the originally blocked multicast request may become successful, which can reduce the amount of blocked multicast requests. Thus, the proposed PFS MRSA algorithm can achieve prominent improvement in service blocking performance when compared with the conventional MRSA algorithms. Simulation results have demonstrated

that the proposed PFS MRSA algorithm has better service blocking performance than the conventional MRSA algorithms.

We organize the remaining part of this manuscript as follows: Section 2 introduces the proposed PFS MRSA algorithm for FGONs; Section 3 presents and analyzes the simulation results of the proposed PFS MRSA algorithm; finally, Section 4 summarizes this paper.

## 3. Partial-Failure Segregated Multicasting Routing and Spectrum Assignment Algorithm for Flex-Grid Optical Networks

In this section, we propose a PFS MRSA algorithm to increase the provisioning efficiency of FGONs via reducing the amount of blocked multicast requests. As the aforementioned conventional MRSA algorithms always consider all or a group of destination nodes of a multicast request as a whole in the path routing and spectrum assignment procedure, they fail to pick out the failure destination leaf-node from a blocked multicast request. Different from them, the proposed PFS MRSA algorithm can segregate one failure destination leaf-node from the originally blocked multicast request and accommodate the failure destination leaf-node and the remained multicast request independently. Since one failure destination leaf-node is isolated from the original multicast request, the remained multicast request can be provisioned with higher success probability. Furthermore, the segregated destination (with the same transmission source as the original multicast request) and the remained multicast request can be accommodated by the network independently. In this way, the originally blocked multicast can be accommodated with higher success probability.

In order to illustrate the proposed PFS MRSA algorithm, an example of the path routing and spectrum assignment in a six-node network is depicted in Figure 1, when the proposed algorithm is employed to accommodate a multicast request. We employ $M$ ($s$, $D$, b) to represent a multicast request, where $s$ is the transmission source node of the request; $D$, consisting of n destination nodes with the expression of $\{d_1, d_2 \ldots d_n\}$, represents the destination set of the request; b indicates the bandwidth requirements (including the guard band) of the request in the unit of FS. In the example, the employed network topology and the multicast-capable node architecture, based on bandwidth-variable wavelength selective switch (BV-WSS), are shown in Figure 1a,b, which illustrates the spectrum utilization on each fiber link in the network before a multicast service, $M_0$ (A, {C, D, F}, 4), arrives. As shown in Figure 1b, the FSs, $FS_5$-$FS_8$, on links A-B, B-C, and C-D are still vacant, while these FSs on other links are already occupied. Therefore, although SPT or MST (i.e., consisting of links A-B, B-C, C-D, and A-F) can be found for all destination nodes (i.e., C, D, and F) in $M_0$, the conventional MRSA algorithms cannot accommodate all the destination nodes successfully when the destination nodes are comprehensively considered as a whole in spectrum allocation. For instance, when the destination nodes, C and D, are successfully accommodated by the FSs, $FS_5$-$FS_8$, along the found SPT or MST, the destination, F, cannot be accommodated successfully by the same FSs. Due to the existence of the failure destination leaf-node, F, the multicast request will be blocked by the conventional MRSA algorithms. By comparison, the proposed PFS MRSA algorithm segregates the failure destination leaf-node, F, from the remaining destination nodes, and allows an independent accommodation for it. Thus, as shown in Figure 1c, the vacant FSs, $FS_1$-$FS_4$, on link A-F, are allocated to $M_0$ for the destination F, while the FSs, $FS_5$-$FS_8$, on links A-B, B-C, and C-D, are allocated to $M_0$ for the remaining destination nodes. In this way, the multicast request $M_0$ can be successfully accommodated, and thus the service blocking performance of the network can be improved. Three key steps are involved in the proposed PFS MRSA algorithm. In the first step, the proposed algorithm tries to seek routing paths and assign optimized spectral resources for a multicast service request. In the second step, the proposed algorithm finds out one failure destination leaf-node of a multicast request when such a request is blocked in the first step. In the third step, the proposed algorithm seeks the routing paths and assigns resources for the found destination and the remaining ones independently. Besides the above illustrative example, pseudo-codes are adopted to

exhibit the details of the proposed PFS MRSA algorithm in Algorithms 1. In addition, a flow chart is depicted in Figure 2 to illustrate the calculation procedures involved in the proposed PFS MRSA algorithm. Noticeably, due to the high-quality components and the advanced digital signal processing technique required by the distance-adaptive resource allocation strategies, we employ a determined not adaptive modulation format (MF) for each kind of multicast request with a comprehensive consideration of both the transmission performance and the spectral efficiency (SE) of each MF.

---

**Algorithms 1: Partial-Failure Segregated (PFS) MRSA Algorithm**

---

1: Pre-compute *K* shortest paths for each node pair in the network
2: **while** network is running **do**
3:    **when** a multicast service request $M$ $(s, \{d_1, d_2 \dots d_n\}, b)$ arrives
4: compute **L** minimum spanning trees (MSTs) for *M* from *s* to all the destination nodes
5:     **for** each MST **do**
6:       **if** *b* contiguous vacant FSs are found along the MST **then**
7:       accommodate *M* by assigning the found *b* FSs along the MST and go to step 30
8:       **end if**
9:     **end for**
10:    **if** *M* is not successfully accommodated along above-computed MSTs **then**
11:      **for** each MST **do**
12:       **for** each destination $d_i$ in the MST **do**
13:       **if** destination $d_i$ is a leaf-node of the MST **then**
14:        segregate $d_i$ from the MST by cutting down its dedicated–correlated link from the MST
15:        **if** *b* contiguous vacant FSs are found along the remained MST **then**
16:         **for** each path *p* in the pre-computed *K* shortest paths for node pair $s$-$d_i$ **do**
17:         seek *b* contiguous FSs along the path *p,* assuming the previously found *b* FSs are already allocated along the remained MST
18:          **if** *b* contiguous vacant FSs are found along the path *p* **then**
19:           assign the found *b* FSs along *p* to accommodate the isolated destination $d_i$; assign the previously found *b* FSs along the remained MST to accommodate other destination nodes and then go to 30
20:           **end if**
21:          **end for**
22:         **end if**
23:        **end if**
24:       **end for**
25:      **end for**
26:     **end if**
27:    **if** no successful spectrum assignment is executed **then**
28:     block *M*
29:    **end if**
30:    update the network
31: **end while**

---

The time complexity of the proposed PFS MRSA algorithm is mainly determined by its adopted strategies in path seeking and spectrum allocation. By adopting the Dijkstra algorithm to seek *K* shortest paths for each node pair and the shortest path matrix [30] and, the path-deleting mechanism [50] to seek *L* MSTs for multicast services, the proposed PFS MRSA algorithm consumes a time complexity of $O((K + L) \times |E| \times (|E| + |V| \log(|V|)) + L \times |E| \times |V|^2)$ in path seeking, with $|E|$ and $|V|$ representing the quantities of the edges and the nodes in the employed network. The complexity consumed by the PFS MRSA algorithm in spectrum allocation is mainly determined by the procedure of seeking available vacant FSs for both the segregated destination and other remaining destination nodes of a multicast request, which can be computed as $O(L \times D_{\max} \times |E| \times |V| + L \times K \times D_{\max} \times |E| \times |F|)$, with $D_{\max}$ denoting the maximum quantity of destination nodes in multicast requests and $|F|$ denoting the total quantity of FSs on one edge. As analyzed above, the time complexity of the proposed PFS MRSA algorithm is polynomial.

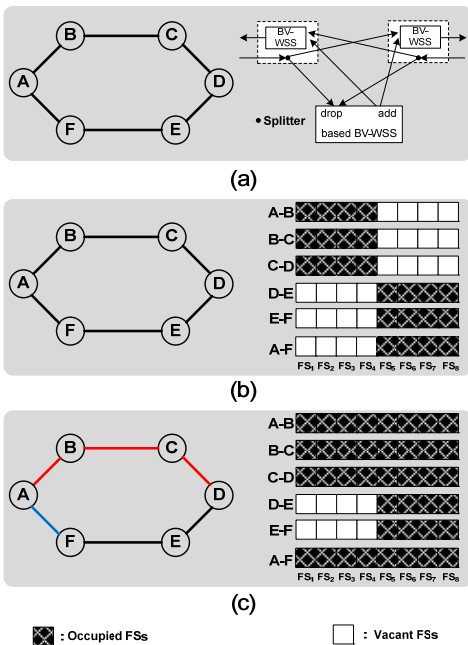

(a)

(b)

(c)

: Occupied FSs          : Vacant FSs

**Figure 1.** An illustrative example for the path routing and spectrum allocation in a six-node network when employing the proposed PFS MRSA algorithm: (**a**) the employed network topology and the node architecture; (**b**) the spectrum utilization on each link before multicast request $M_0$ arrives; (**c**) the path routing and spectrum allocation for $M_0$ when the proposed algorithm is used.

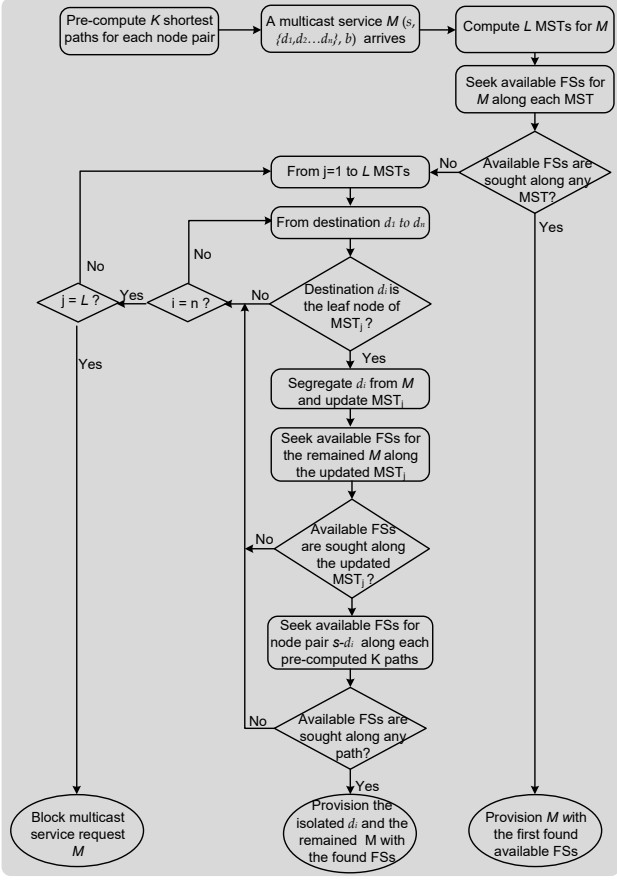

**Figure 2.** Flow chart of the proposed PFS MRSA algorithm.

### 4. The Details of the Proposed PFS MRSA Algorithm

In this section, numerical simulations in two typical networks, NSFNET (in Figure 3) and USNET (in Figure 4), are conducted to evaluate the service blocking performance of the proposed PFS MRSA algorithm via the metric, service blocking probability (SBP), which is calculated according to Equation (1). Considering the high expense and complexity of realizing spectrum conversion in realistic networks, we presume none of the network nodes has such capability in the simulations. In addition, we suppose that one edge is composed of two unidirectional fibers and each fiber contains 4000-GHz spectral resources. By employing 12.5-GHz spectral band as the minimal spectrum-allocation granularity, FS, each fiber link contains totally 320 FSs resources in the simulations. Additionally, due to the industrial needs for high-speed transmission (i.e., beyond 100-Gb/s) and the requirements of the optical transport network (OTN) technology, we assume that three typical types of multicast requests, namely 1-Tb/s, 400-Gb/s, and 100-Gb/s, are accommodated in the simulations [51–53]. By comprehensively considering the transmission performances and the SEs of diverse MFs, we employ 32-QAM, 16-QAM, and QPSK for 1-Tb/s, 400-Gb/s, and 100-Gb/s multicast requests in the simulations, respectively [11,50]. Thus, these three types of multicast requests require (including guard band) 150-GHz [11], 85-GHz [51], and 50-GHz [51] spectral resources, respectively.

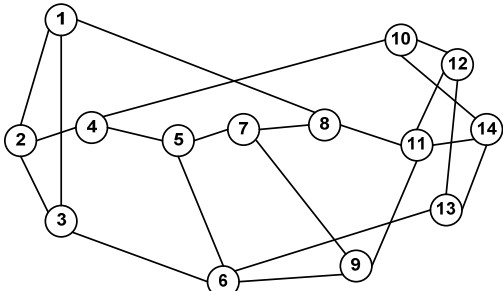

**Figure 3.** 14-node NSFNET.

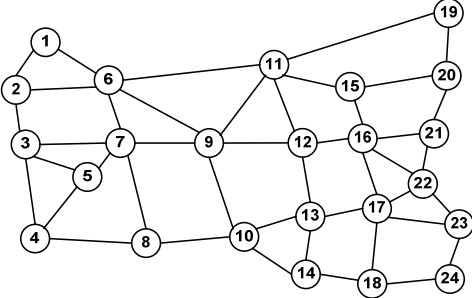

**Figure 4.** 24-node USNET.

In order to simulate dynamic multicast traffic, all multicast requests are randomly generated following a Poisson traffic model with parameter $\lambda$, in which the holding times of the generated multicast requests satisfy a negative exponential distribution with parameter $\mu$. As in [54], the metric $\lambda/\mu$ (in the unit of Erlang) is calculated to indicate the traffic load in our simulations. In the generation of each multicast request, its transmission source and type are randomly determined satisfying uniform distributions, while its transmission destination nodes are also randomly selected from the network nodes with a constant probability, such as 10%, in the simulations. Note that, in order to evaluate the service blocking performance of the proposed PFS MRSA algorithm under diverse conditions, three scenarios with different proportion among the three types of multicast requests are investigated in the simulations.

### 4.1. Scenario One

In scenario one, the proportion of 1-Tb/s, 400-Gb/s, and 100-Gb/s multicast requests is set as 1:1:1 in the simulations. Figure 5 depicts the simulation results on service blocking probability (SBP) in different network architectures under scenario one, when the typical SPT MRSA and the MST MRSA algorithms [30], as well as the proposed PFS MRSA algorithm, are employed. As depicted in Figure 5a, the proposed PFS MRSA algorithm exhibits lower SBP than both the SPT and the MST MRSA algorithms in NSFNET, with either low or high traffic load. For instance, the proposed PFS MRSA algorithm achieves approximately 95% and 79% reductions in SBP when compared with the SPT and the MST MRSA algorithms, respectively, with a low traffic load (such as 80 Erlang). Although these reductions in SBP decrease with the increase in traffic load, the proposed algorithm still gains approximately 30% and 20% reductions in SBP when compared with the SPT and the MST MRSA algorithms, respectively, with the traffic load as high as 200 Erlang. All these indicate that the proposed PFS MRSA algorithm can effectively improve the service blocking performance by isolating the failure destination leaf-node from the original multicast request and accommodating the segregated destination and the remained multicast request independently, especially with low traffic load. This can be understood by the fact that the proposed PFS MRSA algorithm has higher success probability to find available resources to accommodate the segregated destination leaf-node with low traffic load than with high traffic load, since more vacant spectrum resources can be left by lower traffic load. Figure 5b shows the SBP results in USNET under scenario one. Similar to that shown in Figure 5a, the proposed PFS MRSA algorithm gains significant advantage in SBP over both the SPT and the MST MRSA algorithms, although such advantage is more obvious with low traffic load than with high traffic load. With the traffic load as low as 80 Erlang, the proposed algorithm gains about 99% and 96% reductions in SBP than the SPT and the MST MRSA algorithms, respectively. When the traffic load increases to 200 Erlang, the proposed algorithm still realized about 42% and 22% reductions in SBP than the SPT and the MST algorithms, respectively. All the simulation results shown in Figure 5b verify that the proposed PFS MRSA algorithm can effectively enhance the service blocking performance even in a larger network, especially with low traffic load. Noticeably, the difference between the SBPs in Figure 5a,b, may mainly result from the different network architectures and resource distributions in NSFNET and USNET.

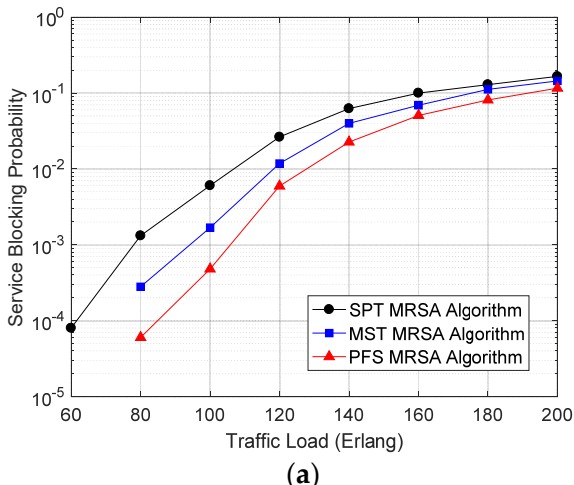
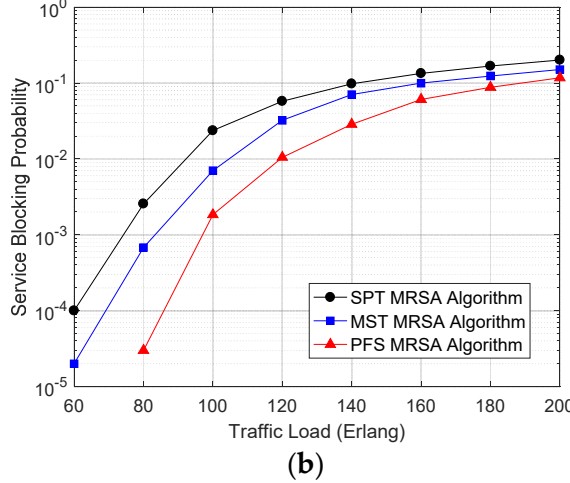

**Figure 5.** Simulation results on service blocking probability in different network architectures under scenario one: (**a**) in NSFNET and (**b**) in USNET.

### 4.2. Scenario Two

In scenario two, the proportion of 1-Tb/s, 400-Gb/s, and 100-Gb/s multicast requests is set as 4:7:12 in the simulations. Figure 6 depicts the simulation results on SBP in different network architectures under scenario two when the SPT and the MST MRSA algorithms [30],

as well as the proposed PFS MRSA algorithm are employed. As depicted in Figure 6a, the proposed PFS MRSA algorithm exhibits the lowest SBP when compared with the other two MRSA algorithms in NSFNET, with different proportions set among the three typical types of multicast requests. This verifies the validity of the proposed PFS MRSA algorithm in enhancing service blocking performance with different proportions among the generated multicast requests, albeit its improvement in service blocking performance decreases with the increase in traffic load. For instance, the proposed PFS MRSA algorithm reduces SBP to zero when the traffic load is less than 80 Erlang, while the other two MRSA algorithms have non-ignorable SBPs with such a low traffic load. As for the high traffic load (such as 200 Erlang), the proposed algorithm still has approximately 65% and 41% reductions in SBP than the SPT and the MST algorithms, respectively. Noticeably, the proposed PFS MRSA algorithm gains more obvious SBP reduction in Figure 6a than in Figure 5a, with either low or high traffic load. This is because the segregated destination of 100-Gb/s multicast service requests has higher probability to find available resources with success than that of 1 Tb/s requests, due to its less bandwidth requirement. Thus, more 100-Gb/s multicast requests in scenario 2 can increase the effectiveness of the proposed algorithm in reducing SBP. Figure 6b shows the SBP results in USNET under scenario two. Similarly, the proposed PFS MRSA algorithm gains remarkable advantage in SBP over the other two MRSA algorithms, although such advantage decreases with the increase in traffic load. As depicted in Figure 6b, the proposed PFS MRSA algorithm obtains 100% reduction in SBP than the other two algorithms with the traffic load less than 80 Erlang. When the traffic load increases to 200 Erlang, the proposed algorithm still gains approximately 57% and 27% reductions in SBP than the SPT and the MST algorithms, respectively. Compared with the simulation results shown in Figure 5b, the results in Figure 6b indicate that the proposed PFS MRSA can enhance the service blocking performance with higher efficiency, even in a larger network, when more low-rate multicast requests (as in scenario 2) are accommodated by the network. All the above results verify the effectiveness of the proposed PFS MRSA algorithm in improving the service blocking performance.

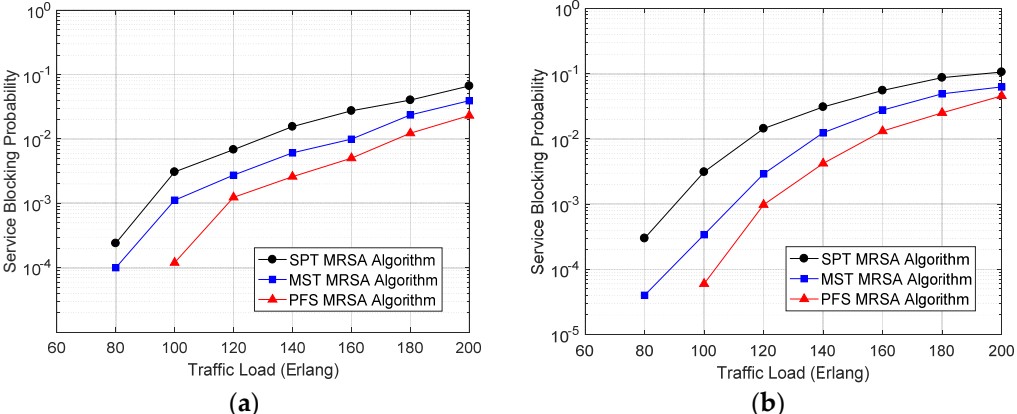

**Figure 6.** Simulation results on service blocking probability in different network architectures under scenario two: (**a**) in NSFNET and (**b**) in USNET.

### 4.3. Scenario Three

In scenario three, the proportion of 1-Tb/s, 400-Gb/s, and 100-Gb/s multicast requests is set as 12:7:4 in the simulations. Figure 7 illustrates the simulation results on SBP in different network architectures under scenario three when the SPT and the MST MRSA algorithms [30], as well as the proposed PFS MRSA algorithm, are employed. As shown in Figure 7a, the proposed PFS MRSA algorithm achieves lower SBP than the other two MRSA algorithms in NSFNET, even with more high-rate multicast requests accommodated by the network. This verifies the effectiveness of the proposed PFS MRSA algorithm in improving service blocking performance with different proportions among the generated multicast requests, although its improvement in service blocking performance decreases with the

increase in traffic load. As shown in Figure 7a, the proposed PFS MRSA algorithm achieves approximately 88% and 73% reductions in SBP than the SPT and the MST algorithms, respectively, with a low traffic load at 80 Erlang. When the traffic load increases to 200 Erlang, the proposed algorithm still gains approximately 18% and 8% reductions in SBP than the SPT and the MST algorithms, respectively. It is worth noting that the proposed PFS MRSA algorithm gains less obvious SBP reduction in Figure 7a than in Figure 5a, with either low or high traffic load. This is because the segregated destination of 1T Gb/s multicast services requests has lower probability to find available resources with success than that of 100-Gb/s requests, and thus more 1 Tb/s multicast requests in scenario three reduce the effectiveness of the proposed algorithm in reducing SBP. Additionally, the SBPs of all the three algorithms decrease due to the existence of more 1 Tb/s multicast requests, which require larger bandwidth and thus have a lower success probability to be accommodated. Figure 7b shows the SBP results in USNET under scenario three. Similarly, the proposed PFS MRSA algorithm gains remarkable advantage in SBP over the other two MRSA algorithms, although such advantage decreases with the increase in traffic load. For instance, when the traffic load is as low as 80 Erlang, the proposed PFS MRSA algorithm obtains approximately 97% and 87% reductions in SBP than the SPT and the MST algorithms, respectively. When the traffic load increases to 200 Erlang, the proposed algorithm still gains approximately 20% and 9% reductions in SBP than the SPT and the MST algorithms, respectively. All these results verify that the proposed PFS MRSA algorithm can effectively improve the service blocking performance in different network architectures, especially with low traffic load.

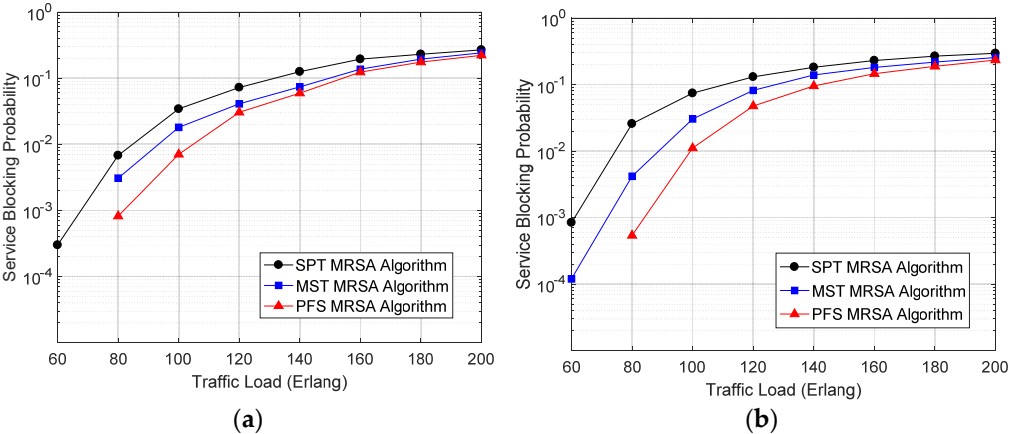

**Figure 7.** Simulation results on service blocking probability in different network architectures under scenario three: (**a**) in NSFNET and (**b**) in USNET.

From the simulation results in the above three different scenarios, we can conclude that the proposed PFS MRSA algorithm gains obvious advantage in reducing SBP over the other two conventional MRSA algorithms. Such an advantage is more conspicuous with low traffic load than with high traffic load. In order to illustrate the effectiveness of the proposed PFS MRSA algorithm in reducing SBP with high traffic load, we collected all the SBP results with the highest traffic load (i.e., 200 Erlang) in Table 1, which indicates that the proposed PFS MRSA algorithm can realize effective SBP reduction even with high traffic load. All the simulation results verify that the proposed PFS MRSA algorithm can effectively reduce SBP by segregating one failure destination leaf-node from a blocked multicast request and accommodating the failure destination leaf-node and the remained multicast request independently.

**Table 1.** Simulation results on service blocking probability in different scenarios when the traffic load is 200 Erlang.

|  | In NSFNET under Scenario One | In NSFNET under Scenario Two | In NSFNET under Scenario Three | In USNET under Scenario One | In USNET under Scenario Two | In USNET under Scenario Three |
| --- | --- | --- | --- | --- | --- | --- |
| SPT MRSA | 0.16548 | 0.06662 | 0.26878 | 0.20104 | 0.10662 | 0.29466 |
| MST MRSA | 0.14414 | 0.03928 | 0.24124 | 0.15058 | 0.06308 | 0.25496 |
| PFS MRSA | 0.11590 | 0.02312 | 0.22161 | 0.11738 | 0.04592 | 0.23324 |

## 5. Summary

In this paper, we propose a partial-failure segregated multicasting routing and spectrum assignment algorithm to improve the service blocking performance in flex-grid optical networks. Different from the conventional MRSA algorithms, which take all the destination nodes of a multicast request into comprehensive consideration during service provisioning, the proposed PFS MRSA algorithm segregates one failure destination leaf-node from a blocked multicast request and accommodates the failure destination leaf-node and the remaining multicast request independently. In this way, the success probability of accommodating the originally blocked multicast request can be increased, which can effectively reduce service blocking probability. Simulation results show that remarkable reduction in service blocking probability can be achieved by the proposed PFS MRSA algorithm, and such reduction even reaches 100% in some scenarios with low traffic load (as in scenario two).

**Funding:** This research was funded by the National Natural Science Foundation of China (Nos. 61971378, 61705190), and the Fundamental Research Funds for the Central Universities, Southwest Minzu University (No. 2021XJTD01).

**Institutional Review Board Statement:** Not applicable.

**Informed Consent Statement:** Not applicable.

**Data Availability Statement:** Not applicable.

**Conflicts of Interest:** The authors declare no conflict of interest.

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
