# Peer review of "Partial-Failure Segregated Spectrum Assignment for Multicast Traffic in Flex-Grid Optical Networks"

_photonics, doi:10.3390/photonics9070488_

Round 1
Reviewer 1 Report
Thank you for this interesting work.
I have some notes that must be considered in the modified manuscript.
------------------------------------------
1) Both (Abstract) and (Conclusion) need some numerical values of (main) results.... to clarify your findings.
2) In your (Introduction), you write:
In order to increase the provisioning efficiency of FGONs, we propose a partial-failure segregated..............
But, I do not see any value of efficiency or its improvement..... (Define the efficiency first)......... May be you mean (service blocking probability) as (efficiency) of system. Please, clarify this point.
3) The procedure of calculations needs to be illustrated (briefly) in a block diagram (or a flow chart).
4) The improvement in service blocking probability is illustrated in figures showing different algorithms for different scenarios.... then ate discussed in text.
I prefer to collect the obtained values for all in a table of comparison to summarize your findings.
5) You used a bit rate 1 Tbps... do you see this value is feasible (practical) or just a theoretical work?
Author Response
Responses to the Comments
We are very grateful to the reviewers for their valuable comments. In the following, we will respond to the comments accordingly.
Reviewer 1:
1. Both (Abstract) and (Conclusion) need some numerical values of (main) results.... to clarify your findings.
Response to reviewer’s comment
As the reviewer’s comment, both abstract and conclusion need some numerical values to address the effectiveness of the proposed PFS-MRSA algorithm in reducing service blocking probability. Given the highest reduction (i.e. 100%) obtained in Scenario II, we addressed such result in both abstract and conclusion to clarify that the proposed PFS-MRSA algorithm can effectively reduce service blocking probability, especially with low traffic load. Thus, in abstract, we added the statement “Simulation results show that the proposed PFS-MRSA algorithm achieves significant reduction in service blocking probability when compared to the conventional MRSA algorithms, and such reduction can even reach 100% in some scenarios with low traffic load.” And in conclusion, we added the statement “Simulation results show that remarkable reduction in service blocking probability can be achieved by the proposed PFS-MRSA algorithm, and such reduction even reaches 100% in some scenarios with low traffic load (as in Scenario 2).”
And we revised the paper accordingly.
2. In your (Introduction), you write:
In order to increase the provisioning efficiency of FGONs, we propose a partial-failure segregated..............
But, I do not see any value of efficiency or its improvement..... (Define the efficiency first)......... May be you mean (service blocking probability) as (efficiency) of system. Please, clarify this point.
Response to reviewer’s comment
As the reviewer’s comment, in order to make the provisioning efficiency more clear, we defined it as the ratio of the successfully-provisioned services to all the generated ones. Thus, it was closely related to service blocking probability, as it can be improved by reducing service blocking probability. Besides, since we focus on minimizing service blocking probability, Pb , to improve provisioning efficiency in this paper, we modeled it using the equation in reference [49] (in the revised version). And we accordingly added the statements “Such provisioning efficiency, defined as the ratio of the successfully-provisioned services to all the generated ones, can be improved by reducing service blocking probability. And we are focus on minimizing service blocking probability, Pb , to improve provisioning efficiency in this paper, which can be formulated as follows [49]:
(1)
where Nb(T) and Nt(T) represent the amounts of the blocked and the total generated multicast service requests in the time duration [0, T], respectively” in the second section of revised paper
3. The procedure of calculations needs to be illustrated (briefly) in a block diagram (or a flow chart)
Response to reviewer’s comment
As the reviewer’s comment, we illustrated the procedure of calculation as following flow chart. We added the flow chart as fig.2 in the revised version and revised the paper accordingly.
4. The improvement in service blocking probability is illustrated in figures showing different algorithms for different scenarios.... then ate discussed in text. I prefer to collect the obtained values for all in a table of comparison to summarize your findings.
Response to reviewer’s comment
As the reviewer’s comment, the figures in all the three scenarios have clearly illustrated that the proposed PFS MRSA algorithm gains obvious advantage in reducing service blocking probability (SBP) over other two conventional MRSA algorithms. And such advantage is more conspicuous with low traffic load than with high traffic load. Therefore, we focus on the SBP results with the highest traffic load (i.e. 200 Erlang), which did not show as obvious SBP advantage as those with low traffic load, to address our findings in the last paragraph of section 4 (in the revised version). Thus, we collected all the SBP results with the highest traffic load (i.e. 200 Erlang) in Tab.2 (in the revised version) to indicate that the proposed PFS MRSA algorithm can realize effective SBP reduction even with high traffic load. And we added the statement “From the simulation results in the above three different scenarios, we can conclude that the proposed PFS MRSA algorithm gains obvious advantage in reducing SBP over other two conventional MRSA algorithms. And such advantage is more conspicuous with low traffic load than with high traffic load. In order to illustrate the effectiveness of the proposed PFS MRSA algorithm in reducing SBP with high traffic load, we collect all the SBP results with the highest traffic load (i.e. 200 Erlang) in Tab.2, which indicates that the proposed PFS MRSA algorithm can realize effective SBP reduction even with high traffic load. All the simulation results verify that the proposed PFS MRSA algorithm can effectively reduce SBP by segregating one failure destination leaf-node from a blocked multicast request and accommodating the failure destination leaf-node and the remained multicast request independently” in the last paragraph of section 4 (in the revised version).
And we added the following table as tab. 2 in the revised version.
|
|
In NSFNET under Scenario 1 |
In NSFNET under Scenario 2 |
In NSFNET under Scenario 3 |
In USNET under Scenario 1 |
In USNET under Scenario 2 |
In USNET under Scenario 3 |
|
SPT MRSA |
0.16548 |
0.06662 |
0.26878 |
0.20104 |
0.10662 |
0.29466 |
|
MST MRSA |
0.14414 |
0.03928 |
0.24124 |
0.15058 |
0.06308 |
0.25496 |
|
PFS MRSA |
0.11590 |
0.02312 |
0.22161 |
0.11738 |
0.04592 |
0.23324 |
Tab.2. Simulation results on service blocking probability in different scenarios when the traffic load is 200 Erlang.
5. You used a bit rate 1 Tbps... do you see this value is feasible (practical) or just a theoretical work?
Response to reviewer’s comment
As the reviewer’s comment, although 1 Tbps transmission has already been realized in lab environments as in [1] (the reference in this response), it is still faced with the difficulties in practical applications. 1 Tbps transmission may still need more research work to be commercialized. Thus, there exists feasible technique to realize 1 Tbps transmission, but we analyzed the 1 Tbps services as theoretical-feasible services.
[1] F. Buchali, K. Schuh, , et al., “1.3-Tb/s single-channel and 50.8-Tb/s WDM transmission over field-deployed fiber”, 45th European Conference on Optical Communication, 2019

Reviewer 2 Report
The paper contains innovative contributions but the following changes/clarifications are needed:
-A Related Work section should be introduced; in particular the authors should described the evolution of the optical networking solutions: Optical Packet [] Switching and Optical Circuit [2] Switching; some references on flexible optical networks should be inserted and the innovative contributions of the manuscript should be highlighted;
-The optimization problem should be at least formulated and its computational complexity should be discussed;
-The choice of the benchmark solutions should be discussed;
-English and writing style should be improved.
[1] V. Eramo, M. Listanti: Wavelength converter sharing in a WDM optical packet switch: dimensioning and performance issues, Computer Networks (Elsevier) vol. 32, no. 5, May 2000, pp. 633-651
[2] L.-W. Chen; E. Modiano, Efficient routing and wavelength assignment for reconfigurable WDM networks with wavelength converters, IEEE Transactions on Networking, Vol. 13, February 2005, pp. 173-186
Author Response
Responses to the Comments
We are very grateful to the reviewers for their valuable comments. In the following, we will respond to the comments accordingly.
Reviewer 2:
1. The paper contains innovative contributions but the following changes/clarifications are needed:
-A Related Work section should be introduced; in particular the authors should described the evolution of the optical networking solutions: Optical Packet [1] Switching and Optical Circuit Switching; some references on flexible optical networks should be inserted and the innovative contributions of the manuscript should be highlighted;
[1] V. Eramo, M. Listanti: Wavelength converter sharing in a WDM optical packet switch: dimensioning and performance issues, Computer Networks (Elsevier) vol. 32, no. 5, May 2000, pp. 633-651
[2] L.-W. Chen; E. Modiano, Efficient routing and wavelength assignment for reconfigurable WDM networks with wavelength converters, IEEE Transactions on Networking, Vol. 13, February 2005, pp. 173-186
Response to reviewer’s comment
As the reviewer’s comment, we revised the introduction section and added a related work section. In the revised first paragraph of introduction, we introduced the WDM optical networks, and three typical optical networking solutions (including optical circuit switching, packet switching, and hybrid circuit/packet switching). And we revised the first paragraph as “In recent years, the intensive growing demands for high-speed services have stimulated the application of wavelength-division-multiplexing (WDM) technology in optical networks, which were known as WDM optical networks [1]. By delivering services via their respective designated wavelengths employing arrayed waveguide grating (AWG), conventional WDM optical networks gained the characteristics of optical circuit switching technique, e.g. consistent bandwidth, order-guaranteed end-to-end delivery [2]. These characteristics were especially suitable for stable traffic requests, but lacked flexibility and intelligence when faced with bursty traffic requests [3]. In order to improve the flexibility and intelligence in processing bursty traffic requests, optical packet switching technology was introduced into WDM optical networks, by employing output buffering technique [4], recirculation buffering technique [5], or shared tunable wavelength converters [6]. Besides, some hybrid packet/circuit switching nodes based on AWGs were designed to realize both packet and circuit switching in optical networks [7]”. In this paragraph, the reference [1] in the above comments was added as the reference [6] in the revised version.
In the second paragraph of introduction, we analyzed the limitation of WDM optical networks, and introduced the characteristics of flex-grid optical networks (FGONs). And we revised the second paragraph as “Although the above optical switching technologies can be adopted by WDM optical networks to improve their flexibility and intelligence when faced with bursty services, but they still had obvious limitations in provisioning heterogeneous services with diverse bandwidth requirements. Since WDM optical networks assigned constant spectral band (e.g. 50 GHz), a.k.a. wavelength, to different services regardless of their respective bandwidth requirements, their spectral allocation flexibility and utilization efficiency were always deteriorated. Although multi-granularity grooming technology was introduced into WDM optical networks to enhance spectral allocation flexibility and utilization efficiency, its spectrum granularity may limit its flexibility [8]. So as to overcome the limitations of WDM optical networks in spectral allocation and utilization, flex-grid optical networks, a.k.a. elastic optical networks, have been proposed [9-14]. By employing finer allocation granularities (i.e. frequency slots) and more tactful routing and spectral allocation algorithms, flex-grid optical networks (FGONs) assigned spectral resources to diverse services dynamically according to their respective bandwidth demands. In this way, FGONs greatly improved the spectral allocation flexibility and utilization efficiency”.
In the third paragraph of introduction, we analyzed the limitation of FGONs in accommodating multicast services. And we revised the third paragraph as “However, the architectures of FGONs are inherited from traditional WDM networks, which may limit the efficiency of FGONs in accommodating multicast services. Such architectures deliver services via dedicated spectral bands and lack mechanisms to take the advantage of the resource-sharing characteristics in multicast transmission. In addition to the architectures, the spectrum continuity and contiguity constraints in resource allocation may further affect efficiency of multicast transmission in FGONs. But the prevalence some novel point-to-multi-point services, e.g. interactive distance learning and high definition IP-TV, make efficient multicast transmission highly desirable in FGONs. And in this paper, we focus on how to reduce the service blocking probability of FGONs in accommodating multicast services, so that more efficient multicast transmission can be realized in FGONs”.
In the added related work section, we reviewed the routing and wavelength assignment algorithms in WDM networks, and some typical RSA algorithms, such as shortest-routing first-fit RSA algorithm [18] (in the revised version), distance-adaptive RSA algorithms [19-20] (in the revised version), and traffic-split RSA algorithms [21-22] (in the revised version), defragmentation-enabled RSA algorithms [23-26] (in the revised version), and group-based RSA algorithms [27-29] (in the revised version). And we added the review statement as “As for FGONs, sophisticated routing and resource assignment (RSA) algorithms are essential to realize their high spectral allocation flexibility and utilization efficiency. Unlike the routing and wavelength assignment (RWA) algorithms adopted in WDM networks [15-17], which seek paths and assign wavelengths for services, RSA algorithms seek paths for service requests and assign optimized number of contiguous frequency slots (FSs) to them [9-10]. By employing the finer spectral allocation granularity, FS (e.g. 12.5 GHz), and allowing different number of contiguous FSs assigned to diverse services, RSA algorithms exhibit higher flexibility and require more management than RWA algorithms. And many RSA algorithms, such as shortest-routing first-fit RSA algorithm [18], distance-adaptive RSA algorithms [19-20], and traffic-split RSA algorithms [21-22], have been proposed for FGONs. In [23-26], some defragmentation-enabled RSA algorithms, employing make-before-break rerouting technique [23], spectrum-retuning technique [24], independent-sets maximizing technique [25], and push-pull technique [26], were proposed to reduce the generated spectral fragments in FGONs. These defragmentation-enabled RSA algorithms improved service blocking performance of FGONs, but might induce traffic disruption or extra system complexity. In order to suppress spectral fragments with low system complexity, group-based RSA algorithms were proposed [27-29]. All the above RSA algorithms were proposed for the unicast (i.e. point-to-point) transmission in FGONs, and lacked efficient mechanisms to accommodate multicast transmission in FGONs” in the first paragraph in the section of related work. In this paragraph, the reference [2] in the above comments was added as the reference [17] in the revised version.
In order to address the innovative contributions of the manuscript, we added the statement “Different from the above conventional MRSA algorithms, in which even one failure destination node will result in a multicast accommodation failure, the proposed PFS MRSA algorithm segregates one failure destination leaf-node from a blocked multicast request before accommodating the failure destination leaf-node and the remained multicast request independently. In this way, the accommodation for the originally-blocked multicast request may become successful, which can reduce the amount of blocked multicast requests” in the third paragraph of related work section.
2. The optimization problem should be at least formulated and its computational complexity should be discussed;
Response to reviewer’s comment
As the reviewer’s comment, we focus on minimizing service blocking probability, Pb, to improve provisioning efficiency in this paper. Thus, we formulated such optimization problem as in [49] (in the revised version):
equation (1) in the revised version.
where Nb(T) and Nt(T) represent the amounts of the blocked and the total generated multicast service requests in the time duration [0, T], respectively. By employing such optimization formulation, we investigated the computational complexity of the proposed PFS MRSA algorithm as “The time complexity of the proposed PFS MRSA algorithm is mainly determined by its adopted strategies in path-seeking and spectrum allocation. By adopting the Dijkstra algorithm to seek K shortest paths for each node-pair, and the shortest-path matrix [30] & the path-deleting mechanism [50] to seek L MSTs for multicast services, the proposed PFS MRSA algorithm consumes a time complexity of O((K+L)×|E|×(|E|+|V|log(|V|)) + L×|E|×|V|2) in path-seeking, with |E| and |V| representing the quantities of the edges and the nodes in the employed network. And the complexity consumed by the PFS MRSA algorithm in spectrum allocation is mainly determined by the procedure of seeking available vacant FSs for both the segregated destination and other remained destination nodes of a multicast request, which can be computed as O(L×Dmax×|E|×|V|+ L×K×Dmax×|E|×|F|), with Dmax denoting the maximum quantity of destination nodes in multicast requests and |F| denoting the total quantity of FSs on one edge. As analyzed above, the time complexity of the proposed PFS MRSA algorithm is polynomial”.
And we revised the paper accordingly.
3. The choice of the benchmark solutions should be discussed;
Response to reviewer’s comment
As the reviewer’s comment, seeking light-trees for multicast service requests is crucial and the first step in provisioning them. And two typical techniques, known as shortest-path-tree (SPT) and minimum-spanning-tree (MST) techniques, can be used to seek light-trees for multicast service requests. Therefore, the SPT and the MST based MRSA algorithms were always adopted as benchmark solutions for MRSA algorithms. And in references [30, 36, 44, 45, 49] (in the revised version), the SPT and the MST based MRSA algorithms were adopted as the benchmark solutions for MRSA algorithm. Thus, we also adopted these MRSA algorithms as the benchmark solutions in performance comparison. And in order to address the choice of the benchmark solutions, we added the statements “Noticeably, as SPT and MST are two typical routing strategies to seek light-trees for multicast services, we adopt these SPT and MST based MRSA algorithms [30] as benchmark solutions to MRSA algorithms in this paper. And we employ these two MRSA algorithms, namely SPT MRSA and MST MRSA algorithms, in the comparison with the proposed PFS MRSA algorithm” in the related work section of the revised paper.
4. English and writing style should be improved.
Response to reviewer’s comment
As the reviewer’s comment, we revised English and writing style of the paper.

Round 2
Reviewer 2 Report
All of my suggestions have been addressed.